# Validation of simulated training sets using a convolutional neural network for isotope identification in urban environments

**Luke Lee-Brewin** *, **Ryan Holden, Caroline Shenton-Taylor**

Physics, University of Surrey, Guildford, Surrey, United Kingdom

* l.j.lee-brewin@surrey.ac.uk

**Data availability statement:** Access to data is controlled by the Nusec SIGMA data challenge

## Abstract

Real-time isotope identification in urban environments can aid law enforcement by providing additional information about the nature of a potential threat. Neural networks have shown promise in isotope identification but the large range of potential isotopes, activities and shielding in uncontrolled urban environments makes creating a representative training set challenging. In this work, a method of generating gamma spectra datasets without requiring radioactive sources is validated with representative data. Simulated spectra are added to background radiation taken from a large dataset of unlabelled gamma spectra (the SIGMA dataset) collected in London by AWE Nuclear Security Technologies. A testing set of 12748 spectra was extracted from the SIGMA dataset by applying k-means clustering to the 10% of spectra with the highest gross counts. Manual inspection and labelling of a subset of each cluster showed that five clusters contained single isotopes and two contained multiple isotopes which were discarded. A convolutional neural network classifier was trained and tested using these two datasets. The model was able to identify isotopes from real SIGMA dataset spectra. The lowest prediction accuracy for a given class was 96% when presented with simulated data, and 89.8% on SIGMA dataset spectra. The high prediction accuracy validates the method for generating spectra and facilitates future work increasing the range of isotopes present in the training set and developing more complex models.

## Introduction

Reliable, real-time isotope identification can aid law enforcement in the interdiction of illicit radiological material within densely populated urban environments. Neural network isotope identifiers are capable of predicting isotopes present in gamma spectra with high prediction accuracies [2–4,6]. Neural networks are a form of supervised learning, and in each case, a dataset of labelled examples is used to train the neural network before deployment. Due to the difficulty in creating a large testing set of gamma spectra in a representative environment, most work in this area has been limited to lab-based or simulated data [2,12,13]. As

(outside of the author's permissions). A citation has been included to request access. The funders had no role in study design, data collection and analysis, decision to publish, or preparation of the manuscript. Below is a statement approved by the authors and all parties involved. The SIGMA dataset is property of the UK Home Office and has been made available to UK academia through the SIGMA Data Challenge, on the conditions that: Data will be deleted from all systems upon the request of UK Home Office. UK Home Office will remain the data owners and uses will not share the data with any other party unless given specific permission. SIGMA data is open but subject to UK Home Office approvals for usage. To apply for access to the SIGMA dataset through the SIGMA Data Challenge, please contact ntr-net@bristol.ac.uk.

**Funding:** This study was partially funded by the Nusec (Nuclear Security Science Network, https://www.nusec.uk/). The funding from Nusec was received by LLB and CST and is associated with the following grant number: UKRI ST/S005684/1. LLB also received a PDRA award from the University of Surrey. The funders had no role in study design, data collection and analysis, decision to publish, or preparation of the manuscript.

**Competing interests:** The authors have declared that no competing interests exist.

the performance of a model is largely dependent on how representative this training dataset is; models are likely to experience a significant drop in prediction accuracy when deployed in new environments.

Spectra collected in densely populated urban environments can differ significantly depending on the exact location they were collected for several reasons [9]. Background radiation is present in all non-shielded spectra and typically contains $^{232}$Th, $^{238}$U, and $^{40}$K which are found in concrete [10,11]. These isotopes will be more abundant in spectra collected in urban environments, and the proximity of a detector to concrete can cause detected background radiation levels to change significantly. The ultimate objective is to deploy this model in an urban environment where the activity of the source, acquisition time, and shielding are all unknown. Creating a representative training set of gamma spectra for isotope identification in uncontrolled environments is a non-trivial task.

In 2018 AWE Nuclear Security Technologies (AWE) collected a large dataset of gamma spectra from sodium iodide (NaI(Tl)) and cesium iodide detectors deployed throughout London. This dataset is an invaluable resource for creating isotope identifiers, spectra collected in-situ across several locations provide a range of background and shielding configurations. Variations in source speed, population density, environmental conditions etc. are all difficult to model on a scale large enough to generate the required gamma spectra for a training set. AWE has made this dataset available to select universities and industry partners as part of the SIGMA data challenge, under which this work is conducted [17].

These detectors were used in three pilot studies and collected 1 s spectra continually for multiple months. Pilot study 3 of the SIGMA dataset contains approximately $7.5 \times 10^8$ spectra collected from over 100 detectors. The subset of pilot study 3 used in this work was created by 15, $2'' \times 4'' \times 16''$ NaI(Tl) detectors placed in key locations around London, including train stations, hospitals, and bridges.

This work explores creating training sets by combining simulated alarms with background radiation. A model was trained to identify 5 isotopes, $^{60}$Co, $^{137}$Cs, $^{133}$Ba, $^{18}$F, and $^{99m}$Tc, all of which are present in the dataset. The performance is then assessed using a testing set of high-count spectra extracted from the dataset and labelled using $k$-means clustering. The model was then shown a real medical alarm consisting of approximately 60 spectra in sequential order with the source passing by a detector. A simple rolling trigger threshold was implemented, activating the model only when the count rate exceeded five standard deviations above the daily average. This average was calculated based on the previous hour of counts, updating once per hour to account for variations in background radiation. However, high-count spectra can skew this average, potentially affecting the trigger's accuracy. Future work could explore more advanced triggering mechanisms to improve reliability in detecting medical alarms.

## Method

The SIGMA dataset includes background spectra and isotopes commonly used in radiotherapy. In this study, the model is designed specifically to identify medical alarms. While isotopes such as $^{232}$Th, $^{238}$U, and $^{40}$K—found in concrete—are present in the SIGMA dataset, their concentrations are too low to be detected in a 1-second spectrum. Therefore, they-and any isotopes not found in the dataset-are excluded from this analysis. The primary goal of this work is to validate a method for generating spectral datasets by testing against real-world representative examples. Future studies could explore the characterisation of these isotopes to better understand variations in background radiation over time.

To identify medical alarms, a neural network must be trained on labelled examples. The SIGMA dataset contains large numbers of medical isotopes in a variety of shielding

configurations but does not contain isotopes that might indicate the presence of a non-medical, uncontrolled radioactive substance which poses a threat to society (threats). To identify threats, a model must be trained on examples. The spectra used to train this model are a hybrid of simulated spectra with added background spectra from the SIGMA dataset. While this work limits the spectra to medical isotopes, by simulating the spectra and validating this approach using real examples; it is likely that this method can be used in future to train a model on simulated threats as well as medical isotopes.

## Training set

Simulated spectra were created using GEANT4, a C++ toolkit for modelling the transport of radiation through matter [7,8]. Radioisotope sources were placed in an air-filled world with a $2'' \times 4'' \times 16''$ detector, and a library of energy depositions from 5 isotopes and either, 0 cm, 3 cm, or 6 cm of aluminium shielding was generated. The library of energy depositions was then sampled randomly and binned to create spectra. Each combination of isotope and shielding comprised 1000 spectra containing counts ranging from 100 to 10000. Figure 1 is a diagram of the creation of a training set.

Nearby structures can alter the distribution of counts in a spectrum as gamma rays scatter off the surfaces. Backscattering is largely dependent on the geometry of the surfaces surrounding a detector. Modelling backscatter requires a large amount of computational power as only a small fraction of incident gamma rays that scatter off surface will deposit

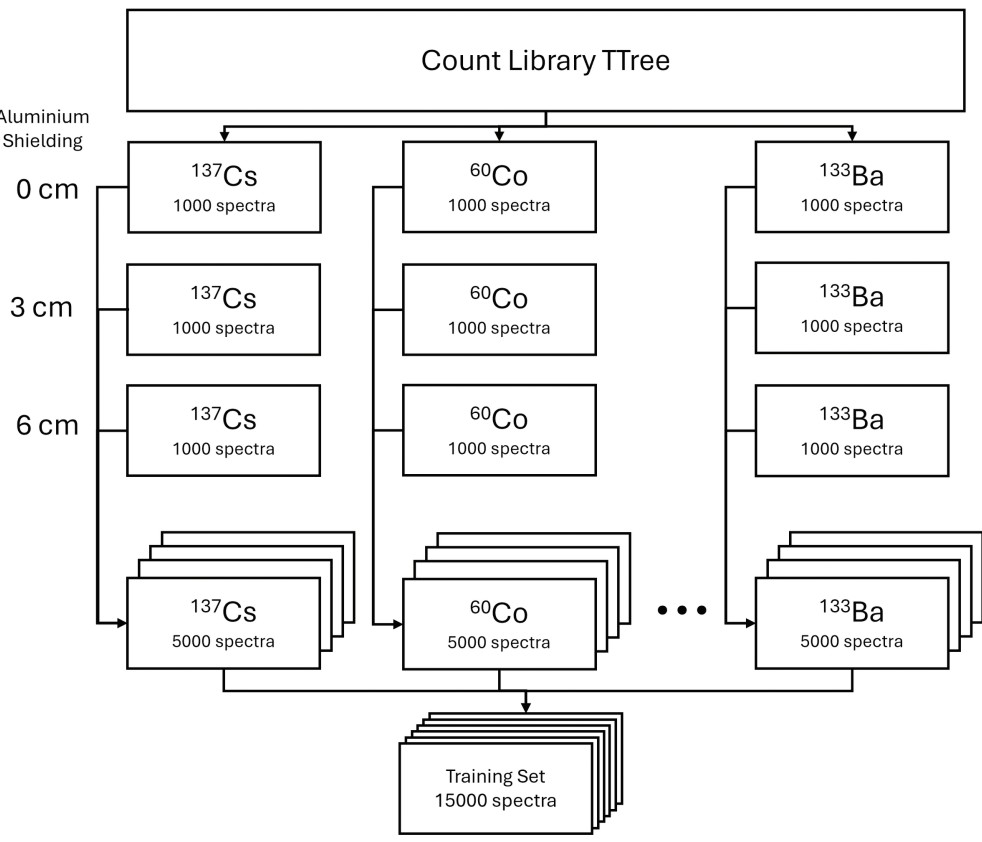

**Fig 1. Breakdown of isotopes and shielding configurations used for training.**

energy into the detector. In this work, computational overhead was minimised by not including a range of environmental geometries to account for backscattering effects.

The calibration of a neural network isotope identifier must be matched to the detectors used to collect the SIGMA dataset. The calibration data provided for the SIGMA dataset is shown in Fig 2 with a linear best fit line.

Detectors deployed outside are likely to experience a change in gain due to temperature fluctuations. Fig 3 shows the average recorded gain values over a 24 hour period against the historical recorded temperature for that day. An detector placed outside was chosen and a day with a large fluctuation in temperature was also selected. This Figure shows a 7% change in

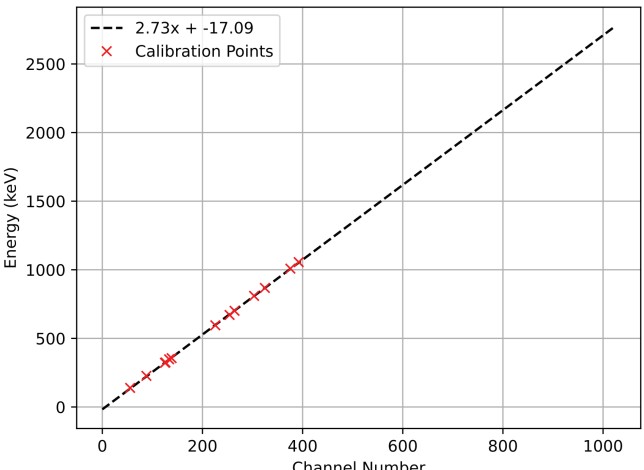

**Fig 2. Calibration data from the SIGMA dataset applied to GEANT4 output.**

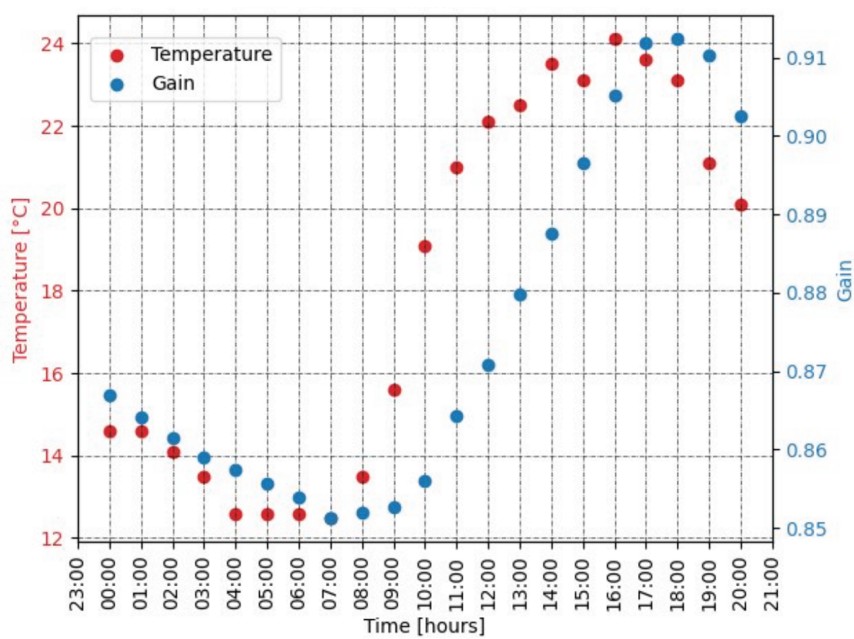

**Fig 3. Gain values and temperature plotted on 01/10/2024.**

the gain over a 12 °C change in temperature change. This value has been rounded to 10% to account for days with more extreme changes in temperature. Each spectrum is then binned with a maximum bin value ± a random calibration factor between 0 and 10% of the maximum bin energy.

Background radiation is then added to the simulated dataset to create a representative training set. The majority of the SIGMA dataset consists of background radiation. Potential alarms were removed from the dataset by removing any spectrum outside of 2 standard deviations ($\sigma$) of the mean number of counts for a given day of data. Between 0 and 3 background spectra from the same detector were added to each simulated spectrum in the training set. Each NaI detector was also used to augment an equal number of simulated spectra.

Since this work focuses on validating the method of generating a simulated training set, minimal preprocessing was applied to the dataset. While it is common to downsample spectra into fewer, potentially non-linear bins to reduce computational overhead, this study retained all 1024 bins to preserve as much spectral information as possible [5]. The only preprocessing step performed was normalisation, ensuring all bin values were scaled between 0 and 1.

## Model

Hypertuning was used to find the optimal structure of the model and a hyperband tuner was to efficiently search the solution space [1]. The number of hidden layers was varied between 2 and 8 during this tuning process as well the number of nodes, and the learning rate.

Two convolutional layers were used to perform feature extraction, the kernal size, stride size and number of features extracted for both of these layers were hypertuned. Additional convolutional layers significantly increase the training and tuning time. As the patterns in gamma spectra are simple geometric shapes *e.g.* a Gaussian photopeak, varying the number of convolutional layers was not deemed necessary. Table 1 shows the hyperparameters varied and the optimal structure obtained by this process.

**Table 1. Range of hyperparameters explored and the optimal structure of the final model.**

| **Convolutional Layer 1** | | | | |
|---|---|---|---|---|
| | **Min.** | **Max.** | **Step Size** | **Optimal** |
| **Filters** | 32 | 128 | 16 | 48 |
| **Strides** | 1 | 5 | 1 | 4 |
| **Kernal Size** | 5 | 50 | 5 | 20 |
| **Convolutional Layer 2** | | | | |
| | **Min.** | **Max.** | **Step Size** | **Optimal** |
| **Filters** | 32 | 128 | 16 | 64 |
| **Strides** | 1 | 5 | 1 | 4 |
| **Kernal Size** | 5 | 50 | 5 | 25 |
| **Hidden Layers** | | | | |
| | **Min.** | **Max.** | **Step Size** | **Optimal** |
| **Number of Layers** | 2 | 8 | 1 | 3 |
| **Layer 1 Nodes** | 32 | 1024 | 32 | 320 |
| **Layer 2 Nodes** | 32 | 1024 | 32 | 384 |
| **Layer 3 Nodes** | 32 | 1024 | 32 | 512 |
| | **Values** | | | **Optimal** |
| **Learning Rate** | 0.1, 0.01, 0.001 | | | 0.001 |
| **Regularisation** | 0.1, 0.01, 0.001 | | | 0.001 |

## Testing set

A subset of the SIGMA dataset spectra containing the highest 10% of counts was used to create the testing set. This subset improved the ratio of alarms to background spectra, and reduced computational overhead. $k$-means clustering was then performed on the dataset to group similar spectra. In $k$-means clustering, $k$ centroids are created representing the number of clusters of data. The Euclidian distance between each of these points and each spectrum in $n$ dimensional space (in this case, $n$ is 1024 for the number of channels in each spectrum). Each spectrum is assigned to the nearest centroid, then the centroids are moved to the mean of all spectra in their assigned cluster. For this work, the centroids were updated 300 times.

To find the optimal number of clusters $k$-means clustering was performed with between 2 and 29 clusters. Fig 4 shows the total distance of all spectra from the nearest centroid as the number of clusters increases. Based on this figure, 8 clusters were chosen for the final clustering. It is typical to choose the number of clusters where the decrease in clustering score slows down significantly, approximately $k = 6$ in this case, a higher value was chosen as a by-eye examination of the clusters was performed to label spectra accordingly.

Random samples of spectra were viewed from each cluster, some clusters were easier to identify as particular isotopes than others. Clusters containing easily identifiable isotopes were then labelled accordingly, and the clusters with multiple isotopes were not included in the testing set. Fig 5 shows this highest count spectrum of each of the clusters kept in the dataset with their associated isotope labelled. $^{137}$Cs and $^{133}$Ba were grouped into the same cluster using this technique. A second $k$-means clustering was performed to differentiate the isotopes.

Perfect accuracy of these labels cannot be guaranteed as the dataset is too large to manually validate all labels. However, it is important to test machine learning models against experimentally collected data in a representative environment. As new data cannot be easily collected in a densely populated area, this method of generating a testing set is a sensible compromise.

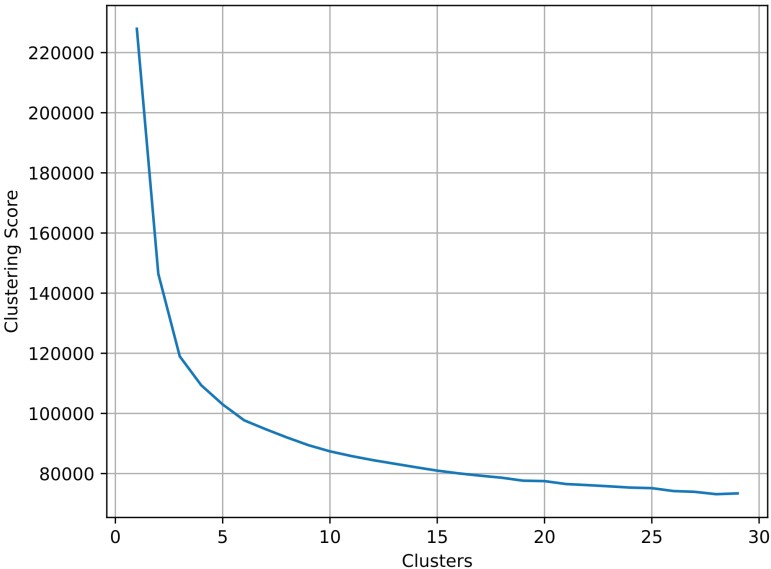

**Fig 4. Clustering score for a high count subset of the SIGMA dataset.**

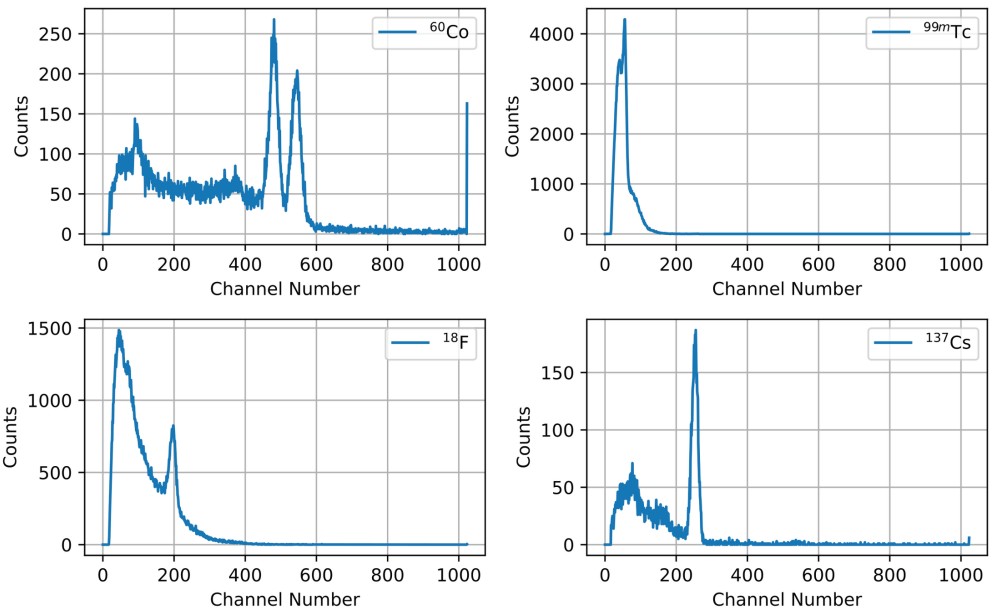

**Fig 5. Highest count spectrum from each cluster with associated isotope manually identified.**

## Results

In an ideal scenario, multiple testing sets should be controlled to assess certain characteristics of a model [2] which can then be used to make informed decisions on future versions of the model. As the intended deployment scenario is a central London, new experimentally collected data cannot be easily created. The scope of this work is limited to assessing the viability of combining representative background radiation to simulated alarms. The performance of the model was therefore assessed against a testing set of real alarms from the SIGMA dataset and the prediction performance of the model from the validation dataset. While this is not ideal, it does provide enough information to explore the viability of creating a training set in this way.

### Prediction accuracy

The model produced a minimum prediction accuracy of 96.0% for $^{99m}$Tc once trained. This prediction performance can be seen in Fig 6 and is reasonably constant throughout the suggesting no bias towards any given isotope.

The testing dataset containing real spectra from the SIGMA dataset produced similarly high prediction accuracies with a minimum prediction accuracy of 89.8% for $^{133}$Ba. Fig 7 shows the prediction accuracies for all isotopes. This testing set was labelled using *k*-means clustering and therefore does not contain equal numbers of each isotope. It was felt increasing the size of this testing set was more important than removing data to equalise the size. Table 2 shows the number of spectra for each isotope class and correct predictions.

### Real-time identification

While the purpose of this work is to explore the real-world performance of a model trained on simulated isotopes, the ultimate goal of this project is to develop a neural network-based,

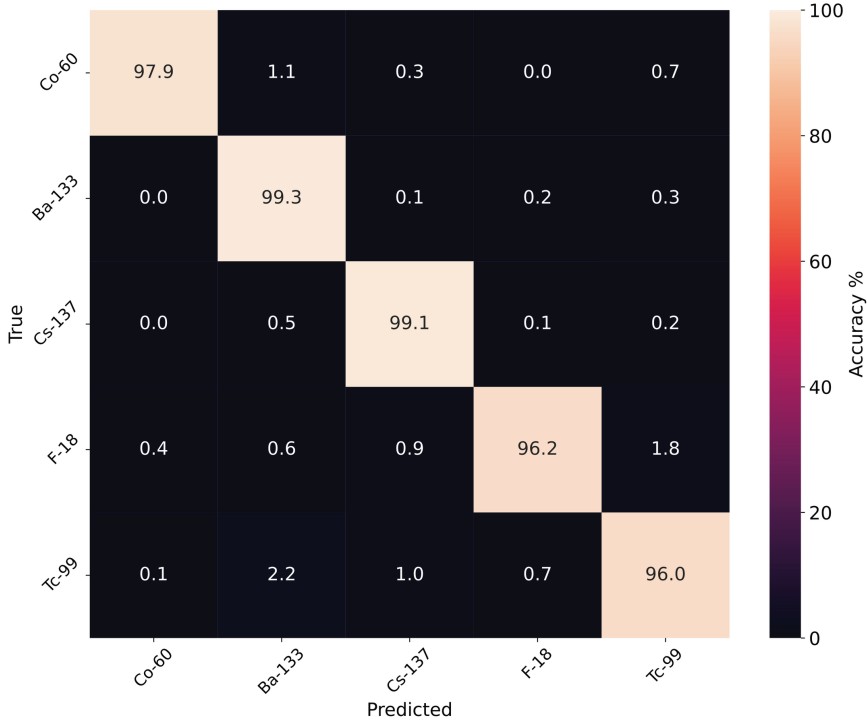

**Fig 6. Confusion matrix showing accuracy of training set predictions.**

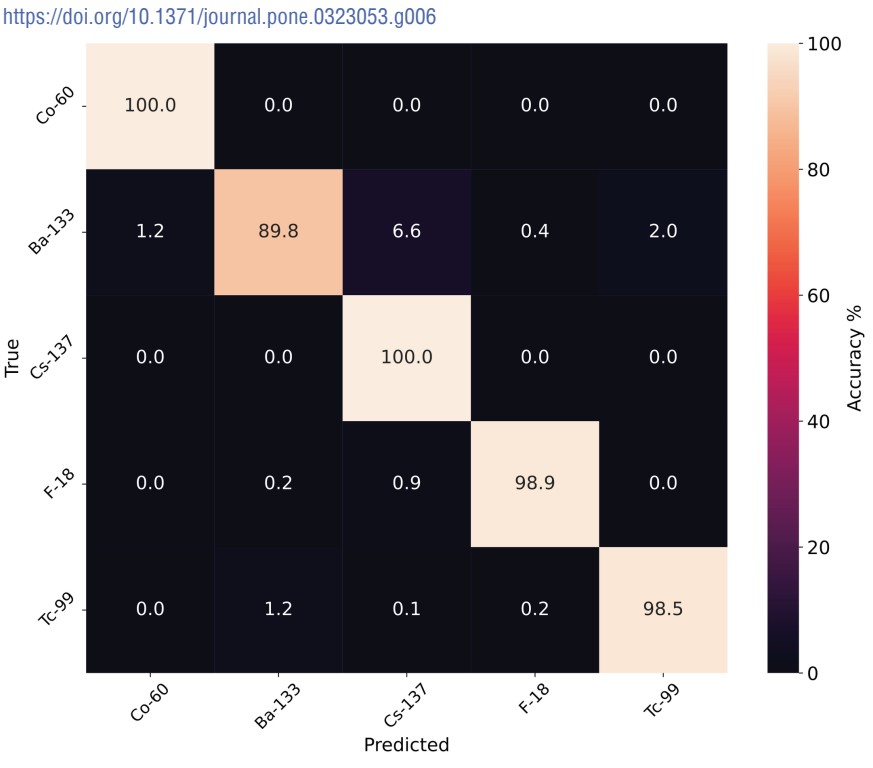

**Fig 7. Confusion matrix showing accuracy of predictions from the SIGMA dataset.**

**Table 2. Number of spectra in each isotope class of the testing set.**

| Isotope | Correct predictions | Class size |
|---|---|---|
| $^{60}$Co | 199 | 199 |
| $^{133}$Ba | 219 | 219 |
| $^{137}$Cs | 100 | 100 |
| $^{18}$F | 6431 | 6503 |
| $^{99m}$Tc | 5643 | 5727 |

real-time isotope identification system. The SIGMA dataset contains a large amount of time-stamped spectra and small subsections of the dataset containing alarms were used to test the performance of the model. To create these small datasets, alarms were identified by finding spectra containing more than 5 standard deviations ($\sigma$) above the average number of counts for a given day. A 30 s window was added to each side of these alarm spectra to capture the source entering and leaving the range of the detector.

Neural networks use large amounts of computational power [14,15]. As alarms are sparsely distributed throughout the SIGMA dataset, more power-efficient methods of identifying an increase in counts are more efficient. This $5\sigma$ approach is simplistic and integrating modern anomaly detection algorithms using unsupervised learning such as principle component analysis or autoencoders are likely to provide more reliable detection of possible threats for further analysis [16].

Fig 8 shows the detection successful identification of $^{18}$F, $^{133}$Ba, and $^{99m}$Tc medical alarms. Sources of $^{60}$Co and $^{137}$Cs were not found using this technique, it is possible that the spectra found in the creation of the testing set were created in controlled detector tests. If the detector was specifically turned on to measure a sealed source, the $5\sigma$ threshold would not work as

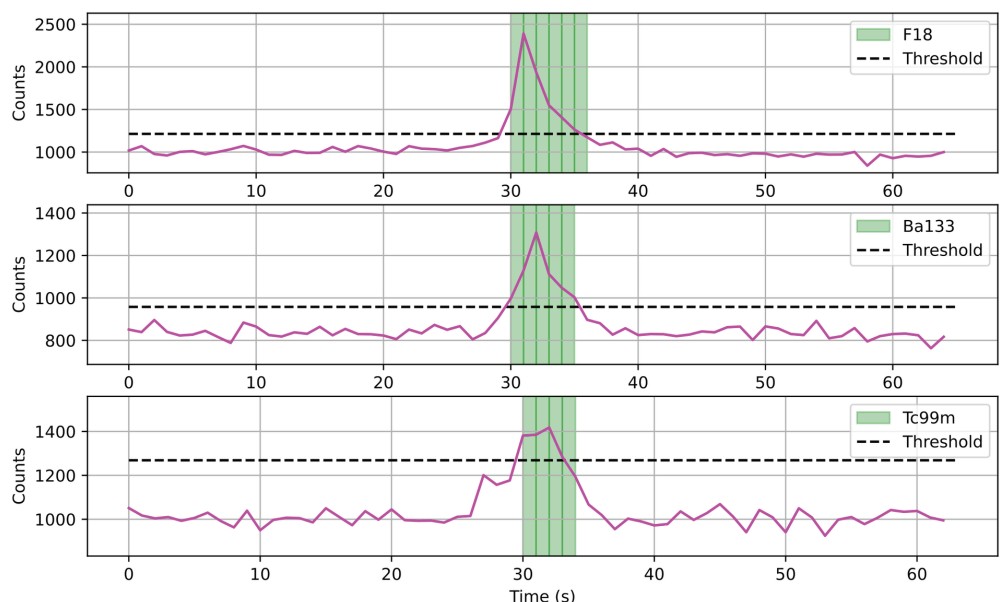

**Fig 8. Total counts detected from alarm start for $^{18}$F, $^{133}$Ba, and $^{99m}$Tc. Green regions indicate correct predictions over the detection threshold.**

there would not be a large number of background spectra collected on that day to lower the average counts detected.

## Conclusion and discussion

This work presents a method to create optimised training sets from large datasets of unlabelled gamma spectra. A testing set of spectra created using the same method as the training set shows a minimum prediction accuracy of 96 % when tested on real SIGMA data.

The creation of the testing set required a manual verification of the presence of isotopes. While every spectrum inspected was found to contain the correct isotope for the assigned k-means cluster, the inspection process was not exhaustive. It remains possible that some spectra were mislabelled and this model's exact accuracy may differ from the presented accuracy. The actual accuracy is not expected to deviate significantly from the presented values as every spectrum inspected was found to be correctly labelled.

A $5\sigma$ trigger threshold was applied to a 24 hour time period. The model was shown small, approximately 60 s datasets containing alarms moving past the detector. In all cases, the model displayed the same high prediction accuracy within seconds of the number of counts exceeding the trigger threshold. The conclusion drawn from these results is that this method of generating gamma spectra can create isotope identifiers capable of performing in real-world scenarios with a high prediction accuracy.

The scope of this work was limited to demonstrating that a model could be created using simulated spectra with added background from the deployment environment and detect radioisotopes in real-time. A model was trained to identify 5 isotopes with unique and distinct signatures. The results presented in this work show that this model can be trained using this method to identify isotopes not present in the SIGMA dataset, such as those with no common medical usage.

Future isotope identifiers intended for deployment will contain a wider range of isotopes and mixtures of isotopes associated with weapons. Spectra in the SIGMA dataset contain time stamps. Training a model to evaluate all spectra collected in a time window will also be explored. Increasing the amount of information in this way may improve prediction performance. Improvements to the simulations exploring efficient methods of incorporating backscattering from surrounding environments are being actively pursued. This could allow individual models to be trained given the known position of a detector in a system and its proximity to nearby structures.

Improvements can be made to this methodology. The *k*-means clustering is a simple clustering method, anomaly detection performed on a different gamma spectrum dataset using Shannon entropy has shown promising results [9]. The trigger threshold is also simple and does not account for a large change in counts due to an electronic anomaly, or a slow drift in count rate due to seasonal temperature change. Further testing focusing on improving these parts of the methodology and benchmarking against existing state-of-the-art isotope identification methods is essential before deploying any machine learning model.

## Acknowledgments

The authors would like to thank NucSec for funding and providing access to the SIGMA dataset, and AWE Nuclear Security Technologies for helpful technical discussions.

## Author contributions

**Conceptualization:** Luke Lee-Brewin.

**Data curation:** Luke Lee-Brewin, Ryan Holden.

**Formal analysis:** Ryan Holden.

**Funding acquisition:** Caroline Shenton-Taylor.

**Methodology:** Luke Lee-Brewin.

**Project administration:** Caroline Shenton-Taylor.

**Software:** Luke Lee-Brewin.

**Supervision:** Caroline Shenton-Taylor.

**Validation:** Luke Lee-Brewin.

**Writing – original draft:** Luke Lee-Brewin.

**Writing – review & editing:** Caroline Shenton-Taylor.

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
