## [Decision Letter · Decision Letter 0]

PONE-D-24-54905

Convolutional neural network isotope identifier created with hybrid simulated and experimental gamma spectra

PLOS ONE

Dear Dr. Lee-Brewin,

Thank you for submitting your manuscript to PLOS ONE. After careful consideration, we feel that it has merit but does not fully meet PLOS ONE’s publication criteria as it currently stands. Therefore, we invite you to submit a revised version of the manuscript that addresses the points raised during the review process.

In this manuscript, Luke et al. present their research on the "Convolutional Neural Network Isotope Identifier Created with Hybrid Simulated and Experimental Gamma Spectra." This research project is both timely and crucial for safety and security considerations, particularly as it advances real-time isotope identification in urban environments. Such capability is essential for providing critical information regarding the nature of potential threats. The authors encounter difficulties associated with isotope identification, given the extensive range of possible isotopes, activities, and shielding factors in uncontrolled urban settings.

The research has generated significant insights, warranting publication in PLOS ONE. The following points support this recommendation:

- First, the research is both original and relevant. The findings hold substantial importance for researchers and regulators within the nuclear security sector. The study could attract considerable interest if the dataset, primarily composed of background radiation with sporadically distributed medical alarms, were expanded beyond the isotopes 18F and 99mTc.

- Second, the research has been conducted with a commendable level of scientific rigor.

Nonetheless, the absence of isotopes such as 40K, 232Th, 226Ra, or U in the background radiation dataset raises questions about the environmental radiation context of the study area and the applied research protocols. It would be beneficial for the authors to address this concern.

Overall, the linearity and clarity of the exposition merit review. The title appears misaligned with the primary objectives articulated throughout the manuscript, including the abstract and the introduction. Additionally, the quality of the article could be enhanced by elucidating the rationale behind the selection of the radioisotopes considered in this research, as well as examining the quality control and quality assurance measures of the presented data.

We look forward to receiving your revised manuscript.

Kind regards,

Cebastien Joel Guembou Shouop, Ph.D., ME, MS

Academic Editor

PLOS ONE

Journal Requirements:

“Initials of Authors: LLB, CST

Partially funded by Nusec (Nuclear Security Science Network)

URL: https://www.nusec.uk/nnsa

LLB: PDRA award 80%, remainder University of Surrey

Data collected as part of the SIGMA data challenged. Formally released to selected academic partners October 2022”

5. We note that you have indicated that there are restrictions to data sharing for this study. PLOS only allows data to be available upon request if there are legal or ethical restrictions on sharing data publicly. For more information on unacceptable data access restrictions, please see http://journals.plos.org/plosone/s/data-availability#loc-unacceptable-data-access-restrictions.

6. Please ensure that you refer to Figure 6 in your text as, if accepted, production will need this reference to link the reader to the figure.

Additional Editor Comments:

In this manuscript, Luke et al. present their research on the "Convolutional Neural Network Isotope Identifier Created with Hybrid Simulated and Experimental Gamma Spectra." This research project is both timely and crucial for safety and security considerations, particularly as it advances real-time isotope identification in urban environments. Such capability is essential for providing critical information regarding the nature of potential threats. The authors encounter difficulties associated with isotope identification, given the extensive range of possible isotopes, activities, and shielding factors in uncontrolled urban settings.

The research has generated significant insights, warranting publication in PLOS ONE. The following points support this recommendation:

- First, the research is both original and relevant. The findings hold substantial importance for researchers and regulators within the nuclear security sector. The study could attract considerable interest if the dataset, primarily composed of background radiation with sporadically distributed medical alarms, were expanded beyond the isotopes 18F and 99mTc.

- Second, the research has been conducted with a commendable level of scientific rigor.

Nonetheless, the absence of isotopes such as 40K, 232Th, 226Ra, or U in the background radiation dataset raises questions about the environmental radiation context of the study area and the applied research protocols. It would be beneficial for the authors to address this concern.

Overall, the linearity and clarity of the exposition merit review. The title appears misaligned with the primary objectives articulated throughout the manuscript, including the abstract and the introduction. Additionally, the quality of the article could be enhanced by elucidating the rationale behind the selection of the radioisotopes considered in this research, as well as examining the quality control and quality assurance measures of the presented data.

Reviewers' comments:

Reviewer's Responses to Questions

**Comments to the Author**

1. Is the manuscript technically sound, and do the data support the conclusions?

Reviewer #1: Partly

Reviewer #2: Yes

Reviewer #3: Yes

2. Has the statistical analysis been performed appropriately and rigorously? 

Reviewer #1: Yes

Reviewer #2: Yes

Reviewer #3: Yes

3. Have the authors made all data underlying the findings in their manuscript fully available?

Reviewer #1: No

Reviewer #2: Yes

Reviewer #3: No

4. Is the manuscript presented in an intelligible fashion and written in standard English?

Reviewer #1: Yes

Reviewer #2: Yes

Reviewer #3: Yes

5. Review Comments to the Author

Reviewer #1: Convolutional neural network isotope identifier created with hybrid simulated and experimental gamma spectra

Here are some constructive comments and questions for the authors:

• The title is not appropriate at all, it should be revised!

• There is no comparison is clear between experimental and simulated results in the Abstract

• The abstract is not appropriate at all, it should be revised and rewritten (Background, methodology, results, conclusion)

• The methodology for combining simulated and experimental gamma spectra is intriguing but needs clearer explanation. For instance, how were discrepancies between simulated and experimental data handled during training?

• While the SIGMA dataset is mentioned, additional details about its composition, diversity of isotopes, and representativeness of real-world scenarios would strengthen the study.

• The study would benefit from benchmarking the CNN against traditional machine learning or other state-of-the-art deep learning models for isotope identification.

• Were any pre-processing techniques applied to the gamma spectra before feeding them into the CNN?

• I recommend to cite the next article that more appropriate for your work (Measurement 168, 108456, 2021)

• What steps were taken to ensure that the CNN generalizes well to unseen datasets?

• The presentation of the work including heading titles should be well revised

These comments should be revised before final decision taken

Reviewer #2: Reviewer’s comments report

First of all, I would like to thank the editor of PLOS ONE for giving this opportunity to revise the manuscript of a entitle of “Convolutional neural network isotope identifier created with hybrid simulated and experimental gamma spectra.”, please kindly find the comments.

General comment:

This manuscript proposes an innovative approach to the identification of radioactive isotopes in complex urban environments, combining real and simulated spectra in order to overcome the lack of representative training data. The database, comprising 750 million spectra collected in London, ensures the requisite realism and applicability. The incorporation of simulated spectra enables the effective management of the inherent variability associated with uncontrolled environments. The convolutional neural network achieves 96% accuracy for medical isotopes such as 18F and 99mTc, thereby demonstrating the effectiveness of the method. The approach is adaptable to non-medical alarms and has strong operational potential for security purposes, offering a high-performance, scalable solution for real-time radiological detection. The paper is well written and all the target points are clear and understood. It may be accepted, after minor revision.

Line 94: “This subset is will contain a higher ratio of alarms to.” The grammatical issue should be correct for more clarity.

Reviewer #3: I understand that the primary purpose of this manuscript is to show a proof of concept for developing an isotope identification system that is trained primarily using simulated data that has been modified to resemble realistic scenarios–such as temperature-dependent gain change, etc.–and real-world background data. The authors then tested the applicability of this network using a subset of the SIGMA dataset. I believe this manuscript contains useful information and should be published, but I have a few concerns and suggestions:

(1) On page 8, the authors state “... more than 5 standard deviations above the average number of counts for a given day”.

What is the method of calculating the average number of counts for a given day? For example, is it a rolling average that continues until the alarm occurs, or is it a total average across the entire day? If it is the latter, how may this method “be effectively deployed to identify isotopes in real time”?

(2) Regarding the Geant4 simulation: Are more details available regarding how the simulation was performed? E.g., was the distance between the source and the detector always stationary? Were there any other obstructions other than the aluminum shielding? Were there any possibilities of gamma ray scattering within the environment?

(3) Would it be possible for the authors to comment on how the manual identification-related errors in the k-means clustering procedure could potentially influence the reported accuracies of the identification network?

(4) It may be beneficial to provide insights on a more impactful method other than the 5-sigma threshold technique for identifying alarms: For example, an unsupervised autoencoder architecture (such as ARAD: https://doi.org/10.1016/j.engappai.2022.104761, among others). I would recommend providing brief information on alternatives and whether or not they would be applicable in this case.

(5) The first sentence of page two states: “ In 2018 the Atomic Weapons Establishment (AWE) created using a subset of the SIGMA dataset.”

I presume this should read “... created a dataset using a subset …”; but, in any case, I believe this sentence should be revised.

6. PLOS authors have the option to publish the peer review history of their article (what does this mean?). If published, this will include your full peer review and any attached files.

Reviewer #1: **Yes: **Mohamed El Tokhy

Reviewer #2: No

Reviewer #3: **Yes: **Randall W. Gladen

---

## [Author Response · Author response to Decision Letter 1]

25 Mar 2025

Response to Reviews

Dear Editor,

Thank you for forwarding the reviewer comments on our manuscript now titled “Validation of simulated training sets using a convolutional neural network for isotope identification in urban environments”. We sincerely appreciate the time and effort the reviewers have taken to evaluate our work and provide constructive feedback. Their insights have been invaluable in improving the quality and clarity of our work.

We have carefully considered each comment and made the necessary revisions. A detailed response to each point is provided in the table below. Unfortunately, we are unable to release the data used for this project as it is held by AWE Nuclear Security Technologies. We have provided a link to request access for this data in the citations.

We appreciate the opportunity to revise our manuscript and look forward to your feedback. Please let us know if any further modifications are needed.

Kind regards,

Luke

University of Surrey

Reviewer 1

The title is not appropriate at all, it should be revised!

The title has been renamed to: Validation of simulated training sets using a convolutional neural network for isotope identification in urban environments

There is no comparison is clear between experimental and simulated results in the Abstract

I have made this clearer by including the lowest class prediction accuracies for the simulated dataset and the experimental dataset.

The abstract is not appropriate at all, it should be revised and rewritten (Background, methodology, results, conclusion)

I have rewritten it to address these concerns.

The methodology for combining simulated and experimental gamma spectra is intriguing but needs clearer explanation. For instance, how were discrepancies between simulated and experimental data handled during training?

The purpose of this work is to generate representative, simulated spectra without requiring new, experimental data. Discrepancies between simulated and experimental data are the focus of this research.

While the SIGMA dataset is mentioned, additional details about its composition, diversity of isotopes, and representativeness of real-world scenarios would strengthen the study. Full analysis of the SIGMA dataset will take years and is currently being undertaken by a group of university and industry partners as part of the SIGMA data challenge.

I have explained more about the SIGMA data challenge and cited the link to request access to the data.

The study would benefit from benchmarking the CNN against traditional machine learning or other state-of-the-art deep learning models for isotope identification.

The validation of this methodology leads onto the testing of larger scale models trained using this methodology. Benchmarking against other isotope identification techniques will be carried out once a larger scale model has been developed. I have addressed this in the conclusion.

Were any pre-processing techniques applied to the gamma spectra before feeding them into the CNN?

The gamma spectra were normalised before being using to train the model. I have emphasised this and explained why no further pre-process was done at this time.

I recommend to cite the next article that more appropriate for your work (Measurement 168, 108456, 2021)

Thank you for highlighting this piece of work, we will benefit from this in our forthcoming research papers.

What steps were taken to ensure that the CNN generalizes well to unseen datasets?

Regularisation was used to avoid overfitting and extensive hypertuning was performed to optimise the structure of the datasets. All testing was performed on experimentally collected data that the model has not been trained on.

The presentation of the work including heading titles should be well revised

I have made the subheadings more concise.

Reviewer 2

Line 94: “This subset is will contain a higher ratio of alarms to.” The grammatical issue should be correct for more clarity.

I have corrected this sentence.

Reviewer 3

On page 8, the authors state “... more than 5 standard deviations above the average number of counts for a given day”.

What is the method of calculating the average number of counts for a given day? For example, is it a rolling average that continues until the alarm occurs, or is it a total average across the entire day? If it is the latter, how may this method “be effectively deployed to identify isotopes in real time”?

I have explained this in more detail including some discussion on the limitations of this methodology.

Regarding the Geant4 simulation: Are more details available regarding how the simulation was performed? E.g., was the distance between the source and the detector always stationary? Were there any other obstructions other than the aluminum shielding? Were there any possibilities of gamma ray scattering within the environment?

I have elaborated on the need to reduce computational overhead as the number of spectra simulated will scale with the number of isotopes newer models are trained to identify. I have also commented on other work being undertaken about modelling backscatter in GEANT4.

Would it be possible for the authors to comment on how the manual identification-related errors in the k-means clustering procedure could potentially influence the reported accuracies of the identification network?

This is an excellent point. I have commented on the prediction accuracy and made it clear that this might alter the results we have presented by a small amount.

It may be beneficial to provide insights on a more impactful method other than the 5-sigma threshold technique for identifying alarms: For example, an unsupervised autoencoder architecture (such as ARAD: https://doi.org/10.1016/j.engappai.2022.104761, among others). I would recommend providing brief information on alternatives and whether or not they would be applicable in this case.

I have cited this paper and commented on other anomaly detection techniques in the conclusion.

The first sentence of page two states: “ In 2018 the Atomic Weapons Establishment (AWE) created using a subset of the SIGMA dataset.” I presume this should read “... created a dataset using a subset …”; but, in any case, I believe this sentence should be revised.

I have corrected this sentence.

---

## [Decision Letter · Decision Letter 1]

Validation of simulated training sets using a convolutional neural network for isotope identification in urban environments

PONE-D-24-54905R1

Dear Dr. Lee-Brewin,

We’re pleased to inform you that your manuscript has been judged scientifically suitable for publication and will be formally accepted for publication once it meets all outstanding technical requirements.

Kind regards,

Cebastien Joel Guembou Shouop, Ph.D., ME, MS

Academic Editor

PLOS ONE

Additional Editor Comments (optional):

As previously mentioned, the quality of the article could be enhanced by elucidating the rationale behind the selection of the radioisotopes considered in this research. But this is left to the author to decide whether to include such details or not.

Reviewers' comments:

Reviewer #1: All comments have been addressed

Reviewer #2: Accept

Reviewer #3: All comments have been addressed

---

## [Editor Report · Acceptance letter]

PONE-D-24-54905R1

PLOS ONE

Dear Dr. Lee-Brewin,

I'm pleased to inform you that your manuscript has been deemed suitable for publication in PLOS ONE. Congratulations! Your manuscript is now being handed over to our production team.

Kind regards,

on behalf of

Dr. Cebastien Joel Guembou Shouop

Academic Editor

PLOS ONE